# An Innovative Approach of Surface Polishing for SRF Cavity Applications

Oleksandr Hryhorenko [1,*], Claire Z. Antoine [2], William Magnin [3], Monish Rajkumar [3], François Brisset [4], Stephane Guilet [5] and David Longuevergne [1]

1 CNRS/IN2P3, IJCLab, Université Paris-Saclay, 91405 Orsay, France
2 CEA, Département des Accélérateurs, de la Cryogénie et du Magnétisme, Université Paris-Saclay, 91191 Gif-sur-Yvette, France
3 LAM PLAN SA, 7 Rue des Jardins, 74240 Gaillard, France
4 CNRS, Institut de Chimie Moléculaire et des Matériaux d'Orsay, Université Paris-Saclay, 91400 Orsay, France
5 CNRS, Institut des NanoSciences de Paris, INSP, Sorbonne Université, 75005 Paris, France
* Correspondence: hryhorenko.lab@gmail.com

**Abstract:** The damage layer produced during the Niobium sheets and cavity fabrication processes is one of the main reasons why cavities have to undergo an extensive surface preparation process to recover optimal superconducting properties. Today, this includes the use of lengthy, costly, and dangerous conventional polishing techniques as buffered chemical polishing (BCP), or electro-polishing (EP). We propose to avoid or at least significantly reduce the use of acids. We developed a novel method based on metallographic polishing of Nb sheets, consisting of 2–3 steps. We demonstrate that this surface processing procedure could be transferred to large dimensions and an industrialized scale thanks to the limited number of steps and its compatibility with standard lapping polishing devices.

**Keywords:** metallographic polishing; superconducting radio-frequency cavity; niobium; surface roughness; damage layer

## 1. Introduction

Superconducting radio-frequency (SRF) cavity is the key element of linear or circular accelerators where radiofrequency (RF) electromagnetic (EM) fields are used for the acceleration of continuous wave (CW) particle beams, see Figure 1 [1,2]. The cavity surface material is subject to intense normal-to-surface electric fields (~100 s of MV/m) and tangential magnetic fields which induce high surface current density (~1–100 s of GA/m$^2$) in the first hundreds of nm, as determined by the cavity geometry. Surface currents on the cavity walls are causing dissipations due to the RF surface resistance (indeed, RF regime superconductors exhibit a non-zero surface resistance, but it is about $10^5$ order of magnitude less than copper). Thus, the crystalline quality of the first hundreds of nm underneath the surface is of paramount importance to ensure optimal superconducting properties. The SRF cavities require a smooth and well-recrystallized surface and need to operate under cryogenic temperatures, typically at superfluid helium temperature, way below the critical temperature of the superconducting material denoted $T_c$ ($T_c$~9.2 K for niobium) [3]. The abrupt transition from superconducting to a normal state, called quench, is not only determined by the critical temperature but also by the amplitude of the magnetic field [4]. Above the critical magnetic field $H_{c1}$, or more specifically, in SRF conditions, above the superheating field $H_s$ [5], the superconductor can be quenched even at cryogenic temperatures [1,3–5]. Nowadays, the preferred material for SRF cavities fabrication is the bulk niobium. This choice is driven not only by its high critical temperature ($T_c$ = 9.2 K) and high magnetic field of first entry $H_{c1}$ (~200 mT) among pure elements, but also by its mechanical properties as a metal; workability and weldability. Other superconducting materials, like Nb$_3$Sn, NbTiN, and MgB$_2$ alloys, have a higher critical temperature but could only be

used as thin films on a workable substrate [6–8]. In both cases, the performance of cavities, evaluated in terms of quality factor $Q_o$ and accelerating field, is significantly impacted by the surface quality in terms of pollution, crystalline structure, and topology [1]. A single microscopic defect such as a microscopic inclusion, scratch, or pull-out becomes a source of local field enhancement and/or heat (normal conducting material) leading to early quench and/or low-quality factors. In order to avoid any reduction of superconducting properties, the surface defects created during niobium sheet fabrication and cavity manufacturing have to be removed to recover a clean and damage-free surface. The so-called damage layer is of the order of 150–200 μm [1,3]. The penetration depth of the electromagnetic field inside the superconducting material is given by the London penetration depth denoted $\lambda_L$. Hence, superconducting properties need to be ensured over several hundreds of nanometers for niobium. Deeper into the material, only good thermal properties are required to ensure efficient heat transport from the RF side to the helium bath side [1,3,5].

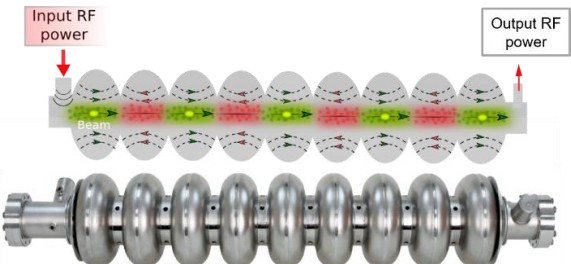

**Figure 1.** TESLA-shaped a 9-cell elliptical cavity made of Nb operating at 1.3 GHz [2].

Standard mechanical polishing can provide very smooth surfaces but a damaged layer (about 100 μm on Nb) is still present at the surface. Buffered chemical polishing (BCP) and electropolishing (EP) are conventional methods used to remove the damaged layer and prepare cavities [9,10]. BCP is composed of acids such as $HNO_3$, $H_3PO_4$, and HF at a typical 1:2:1 ratio. BCP is a relatively simple and fast technique to remove the damaged layer (~4 h). However, BCP is more of an etching process than a polishing process, and as a consequence, the final surface roughness is typically worsened because of the differential between the etching rate versus grain orientation. EP produces smoother surfaces compared to BCP, but this process is more complex (involves the use of an electrode in the mixture of $H_2SO_4$ and HF acids at a 9:1 ratio) and more time-consuming (~8 h). Both methods are very hazardous for operators and exhibit a bad ecological footprint (HF acid). Operational costs of chemical facilities are high because of safety and recycling legislation. Today, these disadvantages become a significant problem because of the increasing demand for SRF units for future large-scale accelerators (FCC, ILC), see Figure 2, which require higher performances, higher reliabilities, and higher repeatability to initiate the construction.

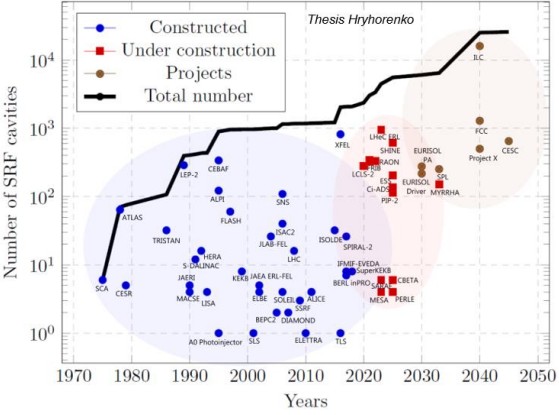

**Figure 2.** Number of SRF cavities versus year [11].

In order to overcome the aforementioned drawbacks, continuous efforts are made in the SRF community to replace or at least significantly reduce the number of acids [12–22]. Since 1995, centrifugal barrel polishing (CBP) has been investigated as an alternative mechanical polishing technique applicable to cavities with a revolution symmetry [16]. The decreased surface roughness after CBP in combination with light EP potentially leads to a higher accelerating gradient and higher achievable quality factor compared to bulk EP [17–20]. However, the extremely long time of treatment (96–150 h), the numerous polishing steps (usually 4–5), and induced mechanical deformations [17], makes CBP not suitable for large-scale production.

In 2009, Calota and al. [23] investigated another type of mechanical polishing, called chemical–mechanical polishing (CMP) as a sub-step of metallographic polishing (MP). They proved that CMP in two steps is successful to obtain smoother surfaces compared to conventional polishing, but pre-treatment of the surface is required. Moreover, this study was limited to small samples.

Our study aimed at going further by developing and optimizing a metallographic polishing procedure transferable to large dimensions (above 10 cm) and integrating a pre-treatment step (lapping). However, contrary to standard chemical polishing and CBP, metallographic polishing can only be applied on flat surfaces. This polishing procedure has thus to be performed directly on niobium sheets before forming, following an alternative path as depicted in Figure 3.

This alternative path can be envisaged as former studies done by Antoine et al. [24,25] showed that the damaged layer is mainly created during the fabrication process of niobium sheets. The fabrication of cavities, involving deep drawing and electron beam welding (EBW), are only responsible for a limited damaged layer of a few micrometers. Further studies are ongoing to consolidate these observations but are not reported here.

This paper is organized as follows. After a brief reminder of the main requirements to produce an SRF-compatible surface, we describe first the alternative metallographic polishing procedure developed on small samples. Second, we show how this procedure has been transferred to large dimensions. Finally, we discuss the results and conclusions.

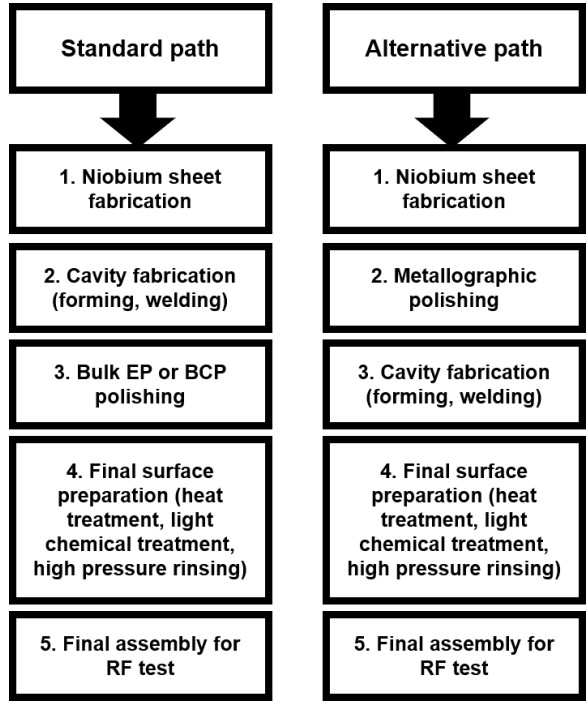

**Figure 3.** Main steps in cavity life cycle for standard and new suggested procedure.

## 2. Materials and Methods

This chapter describes the equipment used in the processing and characterization of small samples and large disks. The metallographic machine MASTERLAM 1.0 was used to polish small samples ($10 \times 10$ mm), whereas the lapping device MM.9100S was used to polish large disks (260 mm in diameter). After the surface processing, weight measurements were conducted using electronic precision balances with an accuracy of 0.001 grams to determine the removed layer. The roughness of the samples was measured using the laser confocal microscope from Keyence (VKX200). The chemical purity of the samples was analyzed using two technologies, secondary ion mass spectrometry (SIMS) manufactured by Hidden Analytical (Compact SIMS-Analyzer), and X-ray photoelectron spectroscopy (XPS) in operation at INSP lab at Sorbonne University. The crystalline quality was studied using a scanning electron microscope (SEM) manufactured by ZEISS (Sigma 300) in combination with the electron backscatter diffraction (EBSD) detector, manufactured by EDAX (Hikari). Finally, hydrogen content was investigated indirectly by triggering on purpose the Q-disease and observing the subsequent residual patterns. Detailed results obtained using this equipment and techniques can be found in the Section 3.

### 2.1. Requirements for SRF Surface Processing

To ensure optimal superconducting properties and to make the surface processing procedure compatible with industrialization constraints, the following requirements can be specified:

- Material removal between 150 and 200 μm to suppress the damaged layer created during niobium sheet fabrication;
- Preserve SRF properties over the first hundreds of nanometers: After the final step, crystalline defects and pollution (embedded abrasives) should be removed;
- To be competitive, the duration of treatment should be comparable to standard chemical techniques (of the order of one working day);
- Limit the number of polishing steps to 2 or 3 s (instead of the 5 or 6 steps typically required in standard metallographic polishing recipes) [25];
- Polishing technique should be transferable to large dimensions.

### 2.2. Metallographic Polishing (MP) Procedure on Small Samples

Flat samples of different dimensions were prepared from a 4.1-mm-thick sheet of high-purity polycrystalline niobium (RRR = 300). Two different types of samples with different initial surface states have been used in this work as depicted in Figure 4. One set of samples was treated by BCP (150 μms of removed material) to ensure that any alteration of the quality (pollution, strain, topology) is only caused by MP surface processing. The second set of samples corresponds to samples "as received". These samples aim at characterizing the efficiency of MP processing to remove the damaged layer created during niobium sheet fabrication. All samples undergo before and after an MP cleaning cycle consisting of the following steps:

- rinsing with deionized water;
- cleaning with deionized water at 30 °C in an ultrasonic bath;
- drying with nitrogen flow;
- cleaning with ethanol.

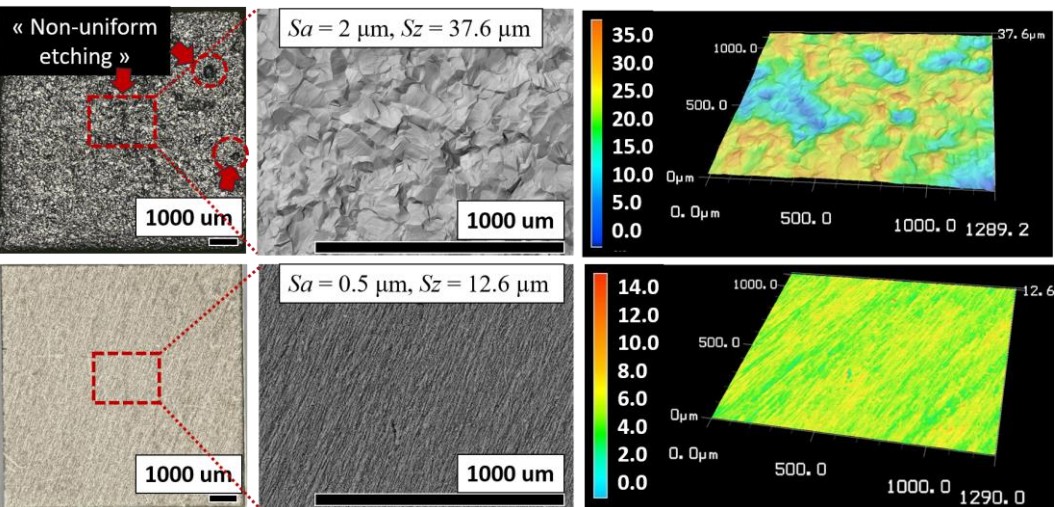

**Figure 4.** Optical and 3D images were acquired by a laser confocal microscope (Keyence VKX200) for BCP-treated samples (**left**) and "as received" samples (**right**). Note: red arrows show variations in surface etching patterns due to hidden structures in the bulk Nb.

Surface processing has been performed on a commercial lapping polishing device MASTERLAM 1.0 manufactured by the French company LAM PLAN. This machine is an automatic device equipped with an oscillating arm pressing the rotating sample holder on the rotating polishing disk support with a diameter of 300 mm giving the possibility to polish samples up to ø150 mm. The polishing parameters such as rotation speeds (disk/sample holder), the direction of rotation (clockwise/counterclockwise), applied pressure, and type and size of abrasives were studied and have been discussed in [11]. The rotational speeds of the sample holder and the polishing disk support were set to 150 rotations per minute (RPM). The lapping/polishing disk and samples were rotated both in the clockwise direction. Moreover, the device was used in an oscillating mode leading to more uniformly distributed abrasives on the samples/disk. In this study, we describe the optimized polishing protocol consisting of two steps. More information on how the process has been optimized can be found in [11].

### 2.3. Metallographic Polishing (MP) Procedure on Large Disks

In order to polish niobium disk dimensions compatible with 1.3 GHz cavity fabrication (260 mm in diameter), the polishing procedure was adapted and optimized for larger lapping devices. For this study, the device MM.9100S was used, fabricated by, and in operation at LAM PLAN company. The large rotating disk with a diameter of one meter and the three independent pistons give the possibility to polish simultaneously three niobium disks. Several issues were faced as device parameters and consumables types differ due to the size difference between metallographic and lapping devices.

## 3. Results and Discussion

### 3.1. Surface Analysis on Small Samples

#### 3.1.1. Step 1: Lapping Step

A rigid composite disk (RCD) made of Cu powder and resin (commercial name CAMEO Gold) in combination with 3 μm polycrystalline diamonds (dia.) is used as a first step. This step aims at planarizing the surface and removing the initial damaged layer caused during fabrication (150–200 μm).

We evaluated the material removal rate (MRR) for different applied pressures (see Figure 5). As expected, the measured MRR is proportional to the applied pressure.

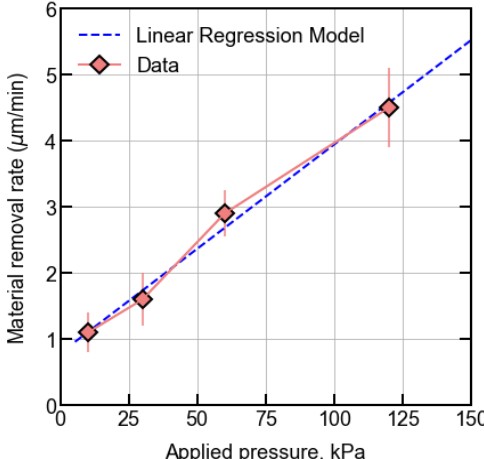

**Figure 5.** Evolution of material removal rate versus applied pressure during the lapping step.

To maintain a constant MRR over time, it is essential to proceed with a continuous supply of abrasives and use disks with a meshed structure acting as an efficient path to remove debris. Hence, as depicted in Figure 6, the damaged layer created during Nb sheets fabrication could be removed within 45–180 min depending on the applied pressure.

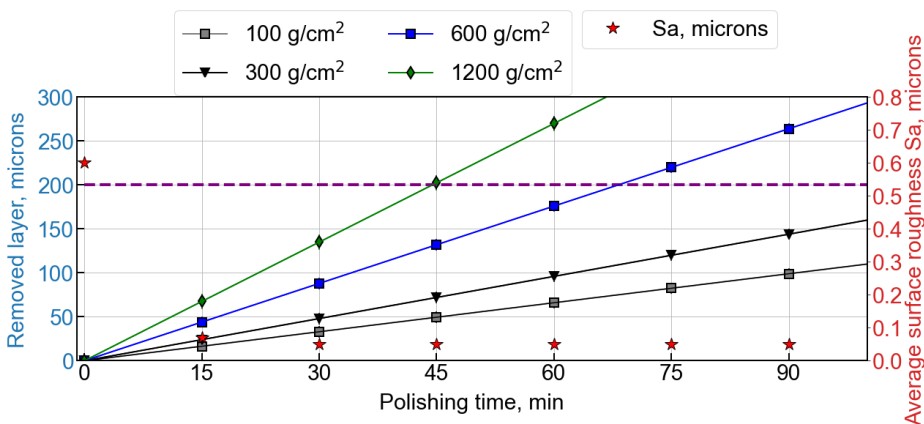

**Figure 6.** Evolution of removed layer and roughness versus time during the lapping step. Note: the left vertical axis applies to solid-line curves, and the right vertical axis (red color) applies to point type (red stars).

After Step 1, surface roughness was examined by a laser confocal microscope. The average surface roughness (Sa) and the maximum height fluctuation of the surface profile (Sz) are measured. After planarization, as depicted in Figure 7, a mirror finish surface with Sa = 50 nm and Sz = 3.11 μm has been obtained.

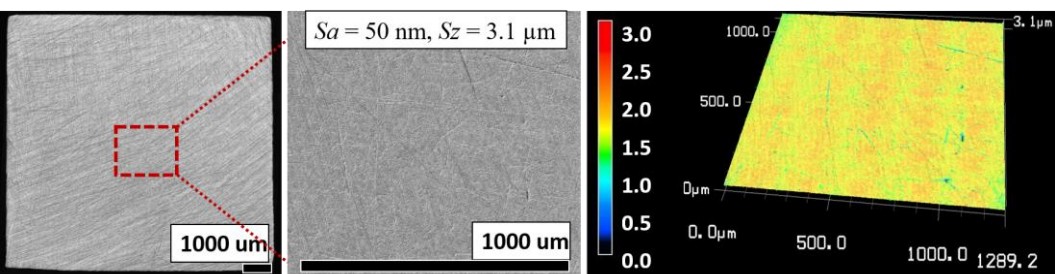

**Figure 7.** Optical images and 3D images acquired by a laser confocal microscope of a Nb sample after step 1.

However, at this stage, after this very aggressive step, the presence of significant crystalline damage and embedded abrasives can be reported. To qualify the depth and density of such pollution and to reveal hidden artifacts, additional processes such as CMP or BCP etching can be performed as surface diagnostics. CMP, as a very soft surface treatment, reveals any surface pollution embedded underneath the surface. As shown in Figure 8 (left), after 15 min of CMP, the surface appears to be strongly polluted by embedded diamonds. On the other hand, BCP tends to reveal crystalline damage because of the differential etching depending on the crystalline orientation and imperfections [11]. As depicted in Figure 8 (right), after a short soaking in BCP solution, the polycrystalline structure is reappearing but with intra-grains patterns, signs of residual strains, and defects [13].

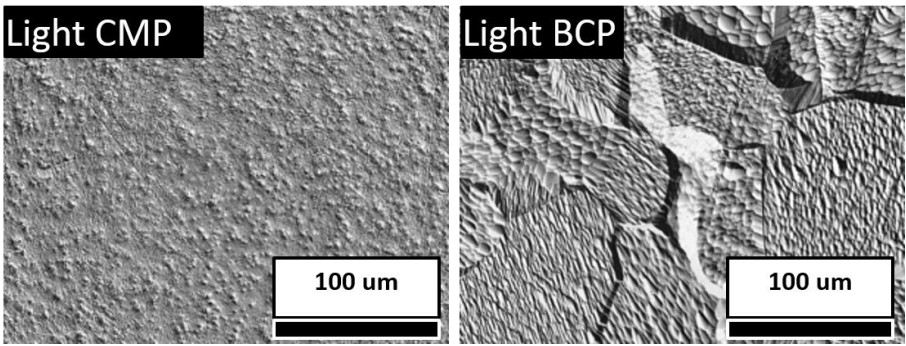

**Figure 8.** Optical images acquired by laser confocal microscope after step 1 followed by a 15-min CMP polishing (**left**), and after a 10-min BCP etching (**right**).

As a conclusion for this first step, the lapping step described appears to be a good compromise between the removal rate and induced pollution and defects. The removal rate significantly exceeds the capabilities of conventional chemical treatment. It makes this treatment very efficient to remove the initial damaged layer. However, this step induces a new damaged/polluted layer which will have to be removed by the subsequent polishing step. A precise measurement of the depth of this induced damaged layer is difficult to perform. Preliminary analysis showed embedded abrasives are within a few micrometers, but sub-grain damages tend to extend over several 10s of micrometers [13,25,26]. An investigation of the residual damaged layer remaining after the final polishing step is presented below.

### 3.1.2. Step 2: Depollution Step

Plain microporous polyurethane cloth (commercial name 4MP1) in combination with 50-nm silica colloidal ($SiO_2$), hydrogen peroxide ($H_2O_2$), and ammonia ($NH_4OH$) diluted in deionized water (up to 20%) is used as a second polishing step. This polishing step, as was mentioned earlier, called CMP, is needed to recover SRF surface properties by removing surface damages (scratches, pull-outs . . . ) and embedded particles caused by Step 1. Step 2 was interrupted every 15 min to investigate surface evolution with a confocal microscope (see Figure 9). Silica colloidal solution is efficient to reduce the amount of embedded abrasives versus time. To describe statistically the number of embedded particles, the edge detection algorithms [27,28] have been applied to the images in Figure 9, and the output of that image recognition is shown in Figure 10. The final surface state of the sample after complete de-pollution is shown in Figure 11. Moreover, after Step 2 the roughness values were obtained using a laser confocal microscope, yielding Sa = 250 nm and Sz = 1.4 μm. It is shown in Figure 12, that to reduce the level of pollution from thousands to a few particles, Step 2 is required to be at least 2 h. However, as can be seen in Figure 9, a long polishing run causes the re-appearance of the grains, which eventually leads to a roughness increase, as depicted in Figure 13. Hence, Step 2 is not efficient in reducing surface roughness, but efficient in improving the micro-structure and chemical purity of the material. To verify the previous statement, the processed surface has to be verified under

RF and at cryogenic temperatures. The first samples were tested at SLAC on a dedicated test bench. Test results have been published in [13] and were not really satisfactory because of Q-disease: the presence of a significant amount of hydrogen in niobium material leads to the formation of niobium hydride precipitates when exposed to low temperatures typically between 50 and 150 K [28]. These niobium hydrides are non-superconducting and lead to a strong degradation of the surface resistance [29–31]. Thus, the triggering of Q-disease on our samples made the evaluation of the impact of the polishing procedure on the surface resistance not relevant.

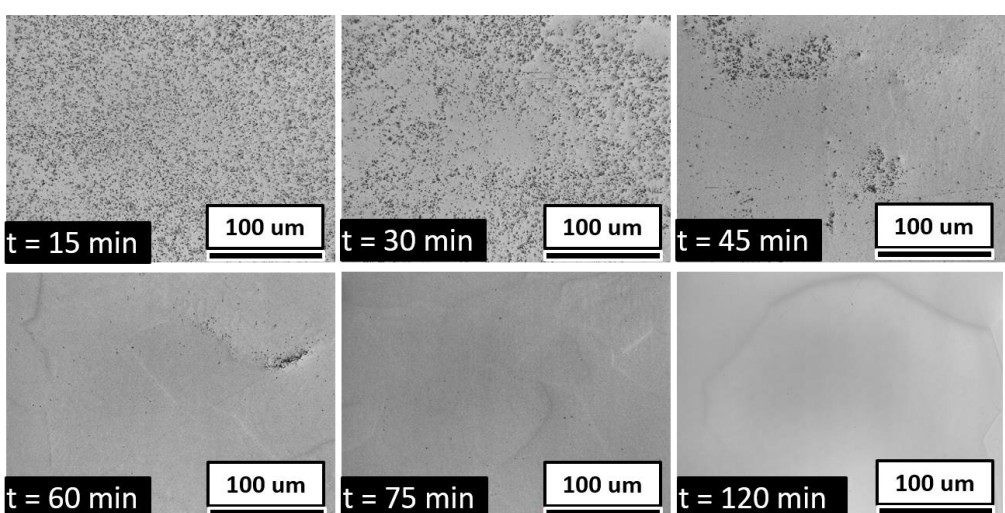

**Figure 9.** Laser confocal images of Nb surface during Step 2 for different processing duration. Note: imagining depicts the pollution removal, grain appearance, and grain growth.

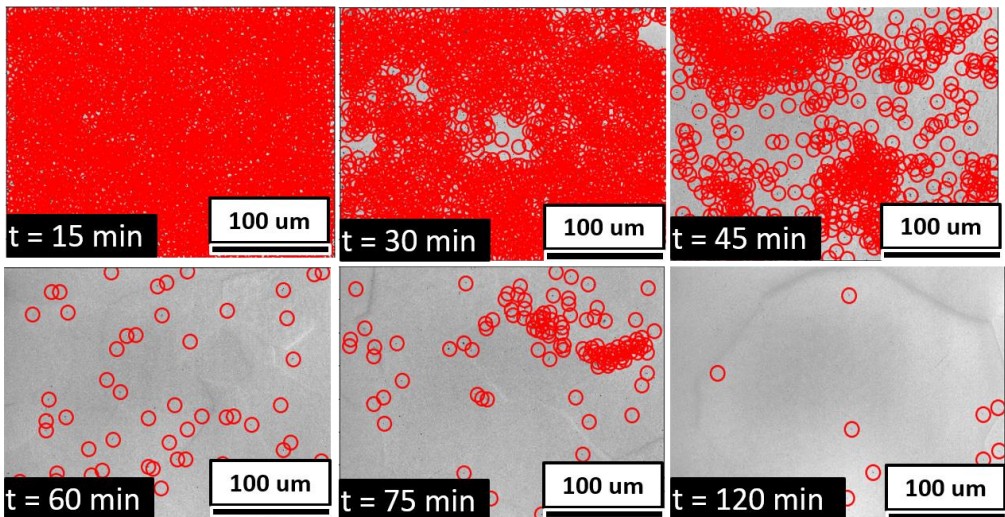

**Figure 10.** Identified particles from the laser confocal images (Figure 9) with edge-detection algorithms (OpenCV). Red circles correspond to locations there are embedded diamonds. Scale is defined in pixels.

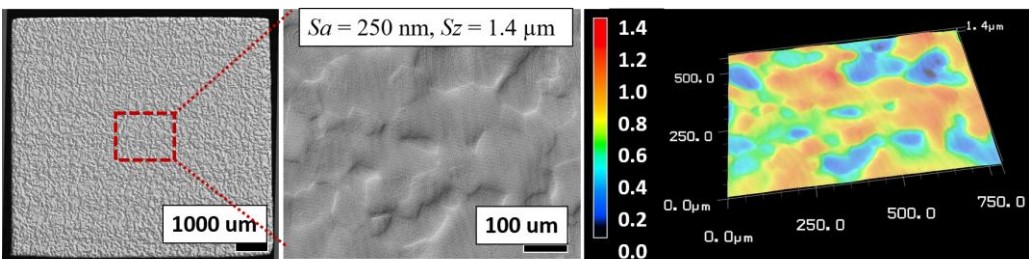

**Figure 11.** Laser confocal images, optical image, and topography of the Nb sample after Step 2.

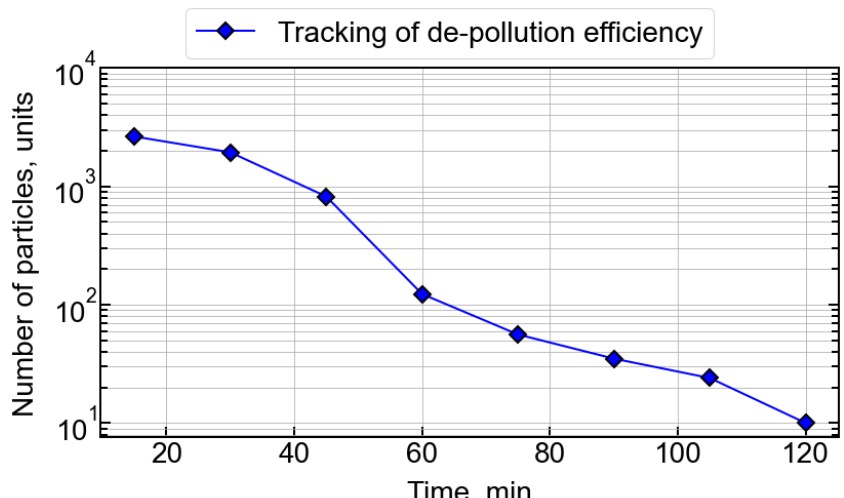

**Figure 12.** The calculated number of embedded particles versus polishing time.

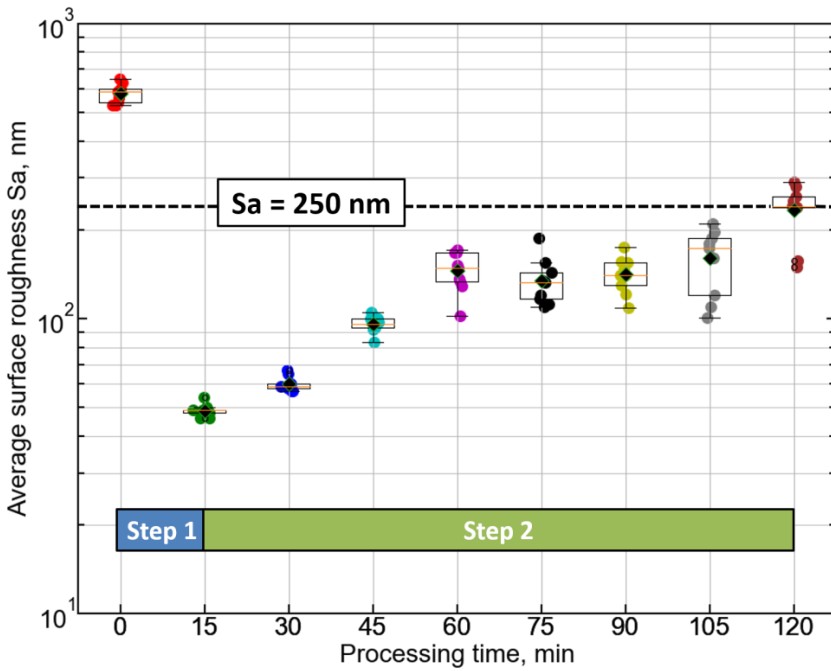

**Figure 13.** Evolution of the average surface roughness as a function of time.

Before producing a new sample for the RF test at a cryogenic temperature additional studies on Q-disease were conducted on the samples after the metallographic procedure. The results of these additional studies are discussed later in this paper.

### 3.2. Final Surface Analysis on Small Samples

As mentioned earlier, the best way to evaluate surface quality after metallographic polishing is to measure surface resistance under radio-frequency waves at cryogenic temperatures. However, this type of test is expensive, technically difficult, and shows a very low availability [32–35]. The final optimization of polishing parameters is not possible by proceeding with an iterative approach using such devices. The use of surface characterization techniques sensitive to pollution and crystalline structure within the penetration depth of the RF wave, meaning a few hundreds of nanometers, is required.

In that sense, the use of a scanning electron microscope (SEM) at low voltage (~2 kV) with standard detectors like secondary electron (SE), back-scatter electron (BSE), and in-lens (SE detector integrated into the SEM column) give a good qualitative assessment of surface pollution and crystalline effects. To address the crystal quality more precisely and quantitatively, advanced analysis like electron back-scatter diffraction (EBSD) has been used. X-ray photo-electron spectroscopy (XPS) and secondary ion mass spectroscopy (SIMS) were also used to assess surface pollution. Finally, hydrogen contamination and mitigation studies were investigated.

#### 3.2.1. EBSD Analysis

EBSD scans of the polished face were performed. The polished surface has been analyzed directly by EBSD without any subsequent surface treatment. As this technique is sensitive to the first 10s of nanometers [36], it makes it relevant to address the quality of the crystalline structure within the RF penetration depth.

As depicted in Figure 14, this 2-step polishing technique does not induce significant crystalline damage. Step 2 is thus efficient and sufficient to remove crystalline damages induced by step 1.

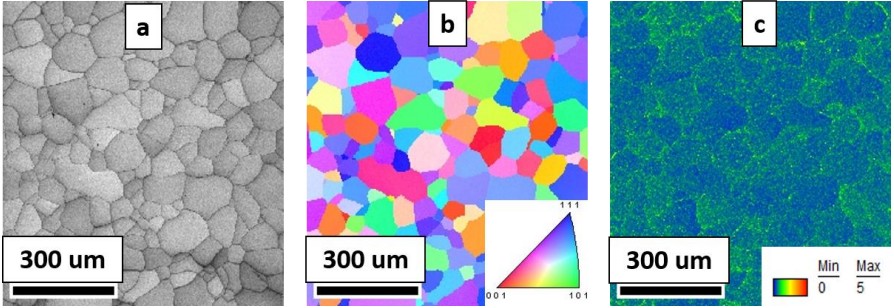

**Figure 14.** Measured EBSD outputs map of the polished face: (**a**) image quality, (**b**) inverse pole figure, (**c**) kernel average misorientation.

To analyze the quality of the microstructure, three maps extracted from EBSD analysis (Figure 14) were used: image quality (IQ), kernel average misorientation (KAM), and inverse pole figure (IPF). IQ, Figure 14a, gives information about any surface features (patterns quality, grain, grain boundaries, eventual defects). IPF image, Figure 14b, shows that samples are polycrystalline, and depict the orientation of each grain. According to this analysis, the fully random grain orientation indicates no distinct texture after polishing (no color is predominant), and the complete diffraction of electrons shows that the polishing procedure doesn't create significant damage. KAM, Figure 14c, gives an estimation of the level of local dislocation density. This value varies from $0°$ to $5°$ in correspondence to color which indicates the density of the dislocations. The color code indicates the following:

- Blue—no dislocation,
- Green—minor dislocation,
- Red—significant dislocation.

As can be seen from Figure 14c, the KAM image indicates that the polished face shows only minor dislocation, showing that the surface is almost strain free.

### 3.2.2. SIMS and XPS Analysis

In an effort to more precisely evaluate the level of surface contamination after the developed metallographic polishing procedure, we used two surface characterization methods, namely SIMS and XPS, and compared results to reference samples.

SIMS stands for secondary electron mass spectrometer and is used for chemical composition evaluation. Analysis was performed on a compact SIMS manufactured by Hidden Analytical company in two different modes: static and dynamic. The static mode gives a global overview of all isotopes between 0 and 200 atomic mass units (amu). The dynamic mode allows measuring the evolution of defined masses in depth. In both cases and in our configuration an ion beam of $Ar^+$ at an energy of 5 kV and at a current of 350 nA is used to sputter an area of $2000 \times 2000$ µm². Sputtered ions are analyzed by a quadrupole analyzer. In such configurations, the mass spectrum was measured and collected during the first minute of analysis, which corresponds to an estimated depth of a few nanometers (the sputtering rate is about 1.5 nm/min).

The measured mass spectrums of the sample after metallographic polishing in comparison to the reference samples (after BCP and EP) are presented in Figure 15. Spectrums have been normalized to the highest peak (Nb) allowing us to quantitatively define the pollution level after different treatments. SIMS analysis of all samples identified the presence of common pollution as listed in Table 1. Above mass 70, see Figure 15 (upper image) the MP spectrum is identical to the EP spectrum. However below mass 70, as highlighted in Figure 15 (bottom image), the MP mass spectrum is showing higher pollution but is limited for several masses such as Si and SiO (due to last polishing step) and stainless-steel compounds. Hence, in order to study the evolution of the species of interest, the depth profiles were acquired and are presented in Appendix A. These show pollution limited to the near-surface within 10 min of analysis corresponding to a few 10s of nanometers.

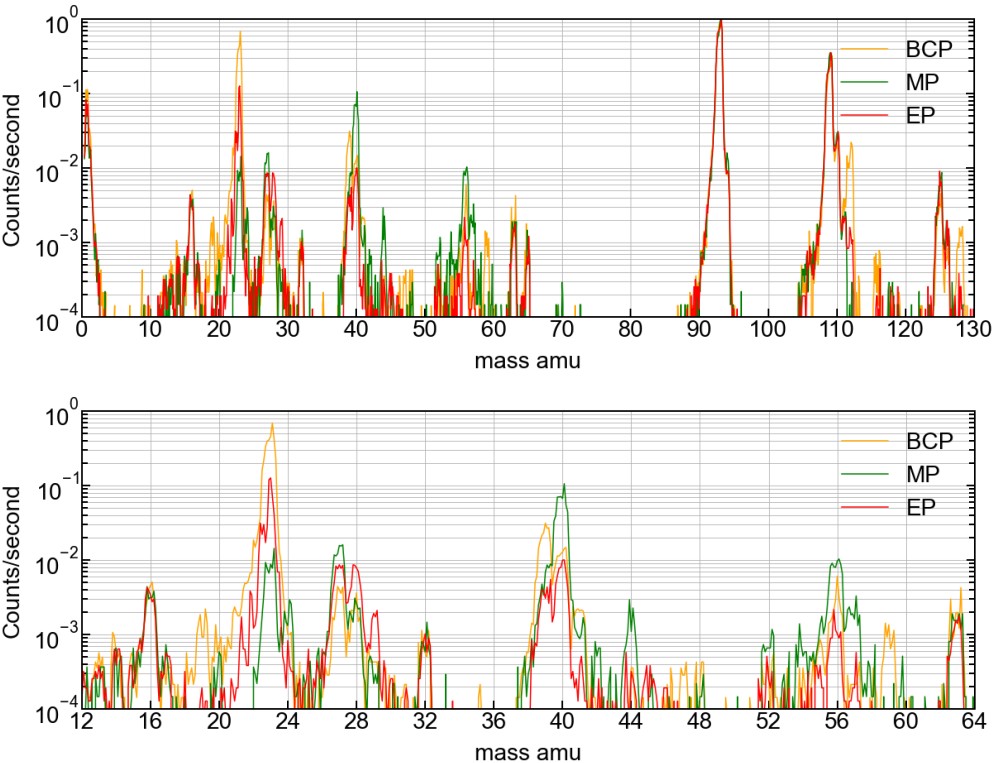

**Figure 15.** Mass spectrum for three types of samples MP (green), BCP (orange), and EP (red). Note: The upper image depicts the entire spectrum in the range of 0–200 amu, whereas the bottom image shows a zoomed-in view of the same spectrum, specifically in the range of 12–64 amu. AMU refers to atomic mass units.

**Table 1.** Main observed Ions and Fragment Species in SIMS spectra recorded on Niobium after different surface processing (MP, BCP, EP).

| Mass, amu | 1 | 14 | 16 | 23 | 27 | 28 | 39 | 40 | 44 | 56 | 63 | 93 | 109 | 125 |
|---|---|---|---|---|---|---|---|---|---|---|---|---|---|---|
| Species | $H^+$ | $CH_2^+$, $N^+$, | $O^+$, $CH_4^+$ | $Na^+$ | $Al^+$, $C_2H_3^+$ | $Si^+$, $CO^+$, $N_2^+$, $C_2H_4^+$ | $K^+$, $C_3H_3^+$ | $Ca^+$, $Ar^+$ | $SiO^+$ | $Fe^+$, $Si_2^+$ | $Cu^+$, $ClSi^+$ | $Nb^+$ | $NbO^+$ | $NbO_2^+$ |

To complete and consolidate the SIMS observation, XPS analysis was used. X-ray photoelectron spectroscopy (XPS) analysis was performed with a test bench at the INSP-CNRS laboratory using a monochromatic Al-K$\alpha$ radiation source at 1486.6 eV. In the analysis chamber, the vacuum was kept at the level of about $10^{-10}$ mbar.

Measured XPS spectrums are presented in Figure 16. The survey spectrums are calibrated to the hydrocarbon impurity C 1s at a binding energy of 284.8 eV. There are visible peaks for O 1s, Nb 3s, Nb 3p1/2, Nb 3p3/2, C 1s, Nb 3d3/2, Nb 5d3/2, Nb 4s, and Nb 4p. The main peak of interest is Nb 3d, for which high-resolution spectrums were measured. Surprisingly, contrary to SIMS analysis, no photoemission signals of Si particles were observed in the spectra.

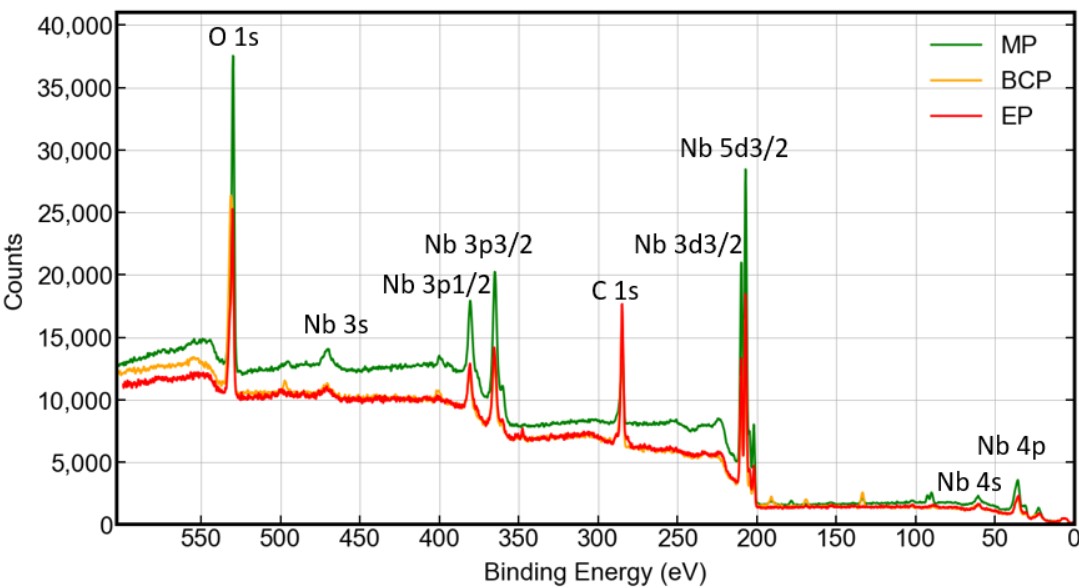

**Figure 16.** XPS survey spectra of Nb samples after MP, BCP, and EP treatment, calibrated with C 1s position at binding energy 284.8 eV.

In order to analyze data and deconvolute the Nb 3d peak, CasaXPS software is used [37]. The reference data of each initial peak position is taken from NIST X-ray Photoelectron Spectroscopy Database [38]. The final binding energy for each peak position and width was determined using the GL (30) function after subtraction of the background using a Shirley method [39,40]. The core-level peak features for metal Nb 3d and Nb 3d oxides were decomposed as doublets with a spin-orbital splitting of 2.7 eV, 2.8 eV, and 2.8 eV for MP, BCP, and EP treated samples respectively. Nb 3d peaks were de-convoluted on the following compounds: $Nb_2O_5$, $NbO_2$, $NbO$, and $NbC$, see Figure 17. The detailed surface composition of each sample (EP, BCP, and MP treated) is presented in Table 2. The composition is presented as a percentage calculated by the CasaXPS software via the multiplication of the peak height at final binding energy by its full width at half maximum (FWHM).

Several statements can be made based on high-resolution XPS spectra. The surface concentration of the pentoxide after MP is reduced from 82.55% (BCP) down to 73.07% (MP), but this thickness would not affect the cavity performance as it is a dielectric layer. However, the increase of other types of oxides will affect the performance, as they might increase the losses. The maximal content of the NbO compound is slightly increased from 6.43% (EP) to 7.6% (MP). At this stage, the oxide layer composition is not primordial as Nb cavities are typically heat treated above 650 °C at the end of the fabrication process to degas the hydrogen and avoid the so-called Q-disease [41].

Finally, MP shows a compatible content of carbon compared to conventional treatment, which confirms that polishing abrasives from Step 1 are efficiently removed, and shows good agreement with optical and SIMS analysis.

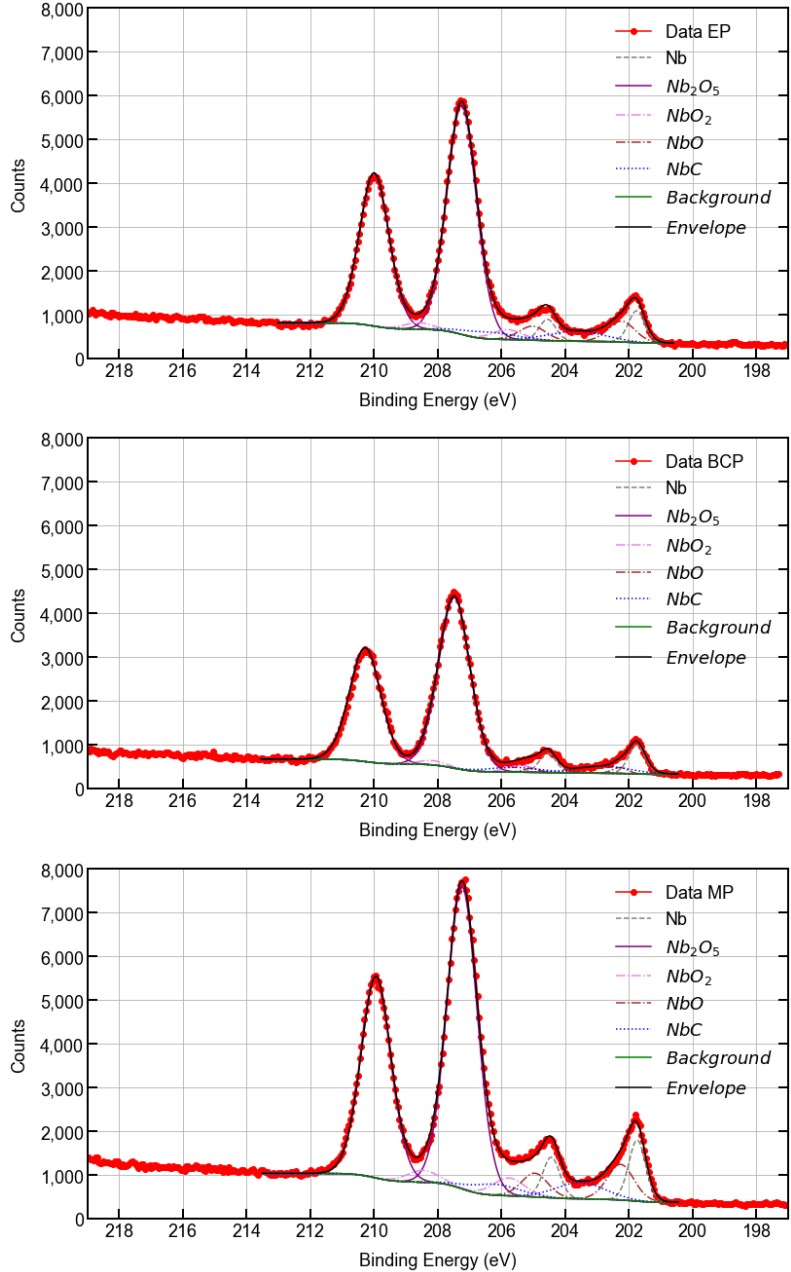

**Figure 17.** High-resolution XPS spectra for the Nb 3d core peak after different treatments (BCP, EP, and MP).

**Table 2.** Composition and chemical states after different polishing treatments (BCP, EP, and MP).

| Polishing | Element | | Composition and Chemical States | | | | |
|---|---|---|---|---|---|---|---|
| BCP | Nb 3d5/2, Nb 3d3/2 | | $Nb_2O_5$ | $NbO_2$ | NbO | Nb | NbC |
| | | Binding energy [eV] | 207.48, 210.28 | 205.39, 208.19 | 202.27, 205.07 | 201.74, 204.54 | 202.79, 205.59 |
| | | FWHM [eV] | 1.16 | 1 | 0.55 | 0.61 | 2.04 |
| | | Composition [%] | 82.55 | 2.75 | 1.35 | 7.38 | 5.97 |
| EP | Nb 3d5/2, Nb 3d3/2 | | $Nb_2O_5$ | $NbO_2$ | NbO | Nb | NbC |
| | | Binding energy [eV] | 207.24, 209.99 | 205.8, 208.6 | 202.22, 205.02 | 201.75, 204.55 | 203.63, 206.43 |
| | | FWHM [eV] | 1.09 | 1.08 | 0.99 | 0.61 | 2.02 |
| | | Composition [%] | 77.82 | 3.31 | 6.43 | 6.03 | 6.4 |
| MP | Nb 3d5/2, Nb 3d3/2 | | $Nb_2O_5$ | $NbO_2$ | NbO | Nb | NbC |
| | | Binding energy [eV] | 207.23, 209.93 | 205.76, 208.46 | 202.25, 204.95 | 201.73, 204.43 | 203.52, 206.22 |
| | | FWHM [eV] | 1.11 | 1.17 | 0.98 | 0.63 | 1.8 |
| | | Composition [%] | 73.07 | 4.51 | 7.6 | 8.38 | 6.43 |

### 3.2.3. Q-Disease Analysis and Mitigation Treatments

Surface treatments such as BCP, EP, and MP cause the pollution of the bulk Nb with hydrogen interstitials [42–44]. This scenario leads to poor cavity performance due to the precipitation of niobium hydrides at low temperatures inducing the creation of irreversible surface defects called "H-footprints". This phenomenon, named Q-disease or 100K-effect in the SRF community, is generally mitigated by degassing hydrogen in a vacuum furnace between 600 and 800 °C. In our case, Q-disease was triggered on purpose to qualitatively evaluate the hydrogen contamination of our samples. A laser confocal microscope was used to observe the creation or absence of H-footprints on the surface before and after 10 h of soaking in liquid nitrogen. This rough method is only sensitive to irreversible crystal damages remaining after warming up in the case of very strong hydrogen contamination. The non-appearance of footprints is not a sign of non-contamination. Moreover, the surface quality must be good enough to be able to observe the H-footprints. Indeed, after the first polishing step, no footprints could be observed although niobium is known to be highly contaminated after a lapping process.

Figure 18 shows the results of the Q-disease tests and mitigation studies after MP treatment. After the full MP process, a significant density of H-footprints was observed on the surface, see Figure 18a, proof of strong hydrogen contamination. Footprints were formed along all analyzed areas (Figure 18a), but their intensity was higher in proximity to pre-existing defects, such as scratches, or higher strains. Moreover, this contamination could be due to the polishing process or even previous steps (niobium fabrication, cutting). In an effort to determine if the last polishing step is responsible for hydrogen loading, the sample was degassed at 600 °C for 10 h under vacuum, see Figure 18b, and applied an additional "Step-2" polishing lasting 30 min (Figure 18c) and 120 min (Figure 18d). No H-footprints are visible after heat treatment nor after additional mechanical polishing with colloidal silica.

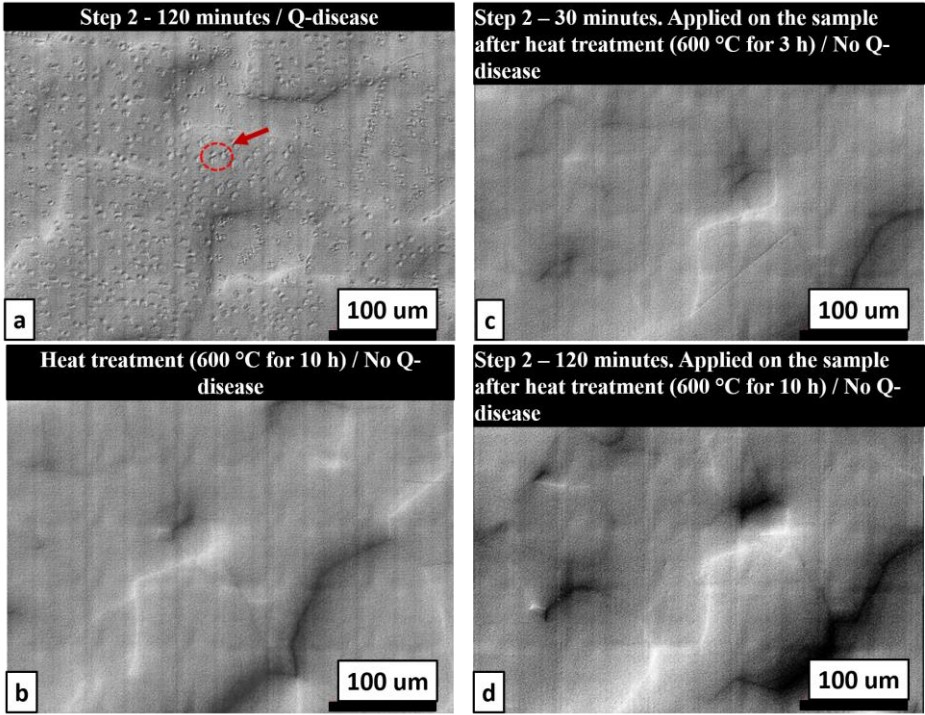

**Figure 18.** Photographs of niobium surface after nitrogen soaking after a full MP process (**a**), after heat treatment (**b**), and additional step-2 polishing (**c**,**d**). Note: Footprints are only visible on the first image. Arrow indicating the red circle is an example of a hydride footprint.

### 3.3. Transfer to Large Dimensions

Firstly, it was impossible to apply the same pressure on samples and large disks during Step 1. As a consequence, to compensate for the reduction of MRR, the grain size of diamonds had to be increased from 3 μm to 6 or 9 μm. Secondly, as rigid composite disks used for small samples are not available for lapping devices, an alternative and equivalent lapping disk had to be identified. There were no alterations made during Step 2.

#### 3.3.1. Optimization of Step-1 for Large Disks

The following types of disks were tested: cast iron, synthetic fiber, and patented disk (LAM PLAN NEW LAM M'M' GREEN), see Figure 19. Cast iron in combination with SiC (Figure 19a) is widely used in industrial applications because of very efficient stock removal, but in the case of Nb, the material tends to be heavily scratched, see in sub-Figure 19b. The synthetic fiber in combination with dia. of 9 μm (Figure 19c) shows good results in terms of final roughness (Sa < 1 μm), but not sufficient enough in terms of material removal rate as the surface quality was not uniform even after 1 h as shown in sub-Figure 19d. Finally, the patented disk LAM PLAN NEW LAM M'M' GREEN (Figure 19e), integrating a meshed structure made of soft and hard material, shows homogeneous polishing over the whole surface, with an acceptable removal rate. The disk could be used either with diamonds of 6 μm or 9 μm. The resulting surface is presented in Figure 19f. Table 3 summarizes the different MRRs depending on the applied pressure and polished area. The measured MRR data is presented here for the lapping machine, as well as for the metallographic device.

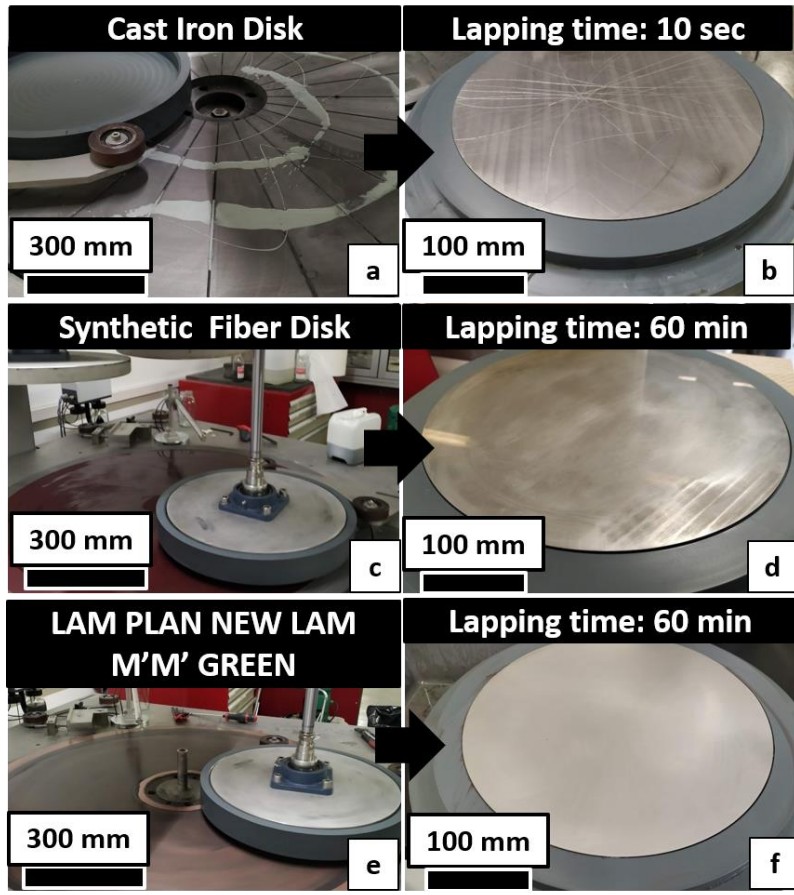

**Figure 19.** Photographs of the tested lapping disks (**a**,**c**,**e**) and surface quality of niobium disks after Step 1 (**b**,**d**,**f**).

**Table 3.** Material removal rate for different polishing configurations.

| Area, cm$^2$ | Pressure, g/cm$^2$ | Disk Type + Diamonds, μm | MRR, μm/min |
|:---:|:---:|:---:|:---:|
| 1 | 1200 | CAMEO Gold + 3 | 4.5 ± 0.6 |
| 1 | 600 | CAMEO Gold + 3 | 2.9 ± 0.4 |
| 1 | 300 | CAMEO Gold + 3 | 1.6 ± 0.4 |
| 1 | 200 | CAMEO Gold + 3 | 1.1 ± 0.3 |
| 125 | 100 | CAMEO Gold + 3 | 0.6 ± 0.1 |
| 530 | 200 | New Lam M'M' Green + 6 | 1 ± 0.1 |
| 530 | 200 | New Lam M'M' Green + 9 | 1.5 ± 0.1 |

### 3.3.2. Observed Issue during Step-2 for Large Disks

The main issue observed for this polishing step was the non-uniformity of surface quality when using a standard plain polishing pad. Indeed, the apparition of two types of regions (defined as "bright" and "dark") was observed after one hour of polishing. Laser confocal observations of "bright" spots, see Figure 20, revealed a large concentration of embedded diamonds, whereas "dark" spots showed very limited contamination. This problem was solved, as in a previous study for RF disks of 13 cm diameter [11], by using a meshed polishing pad. This non-uniformity during the de-pollution process was explained by a non-optimal distribution of liquid over the polishing pad. The final surface results are presented in the following section.

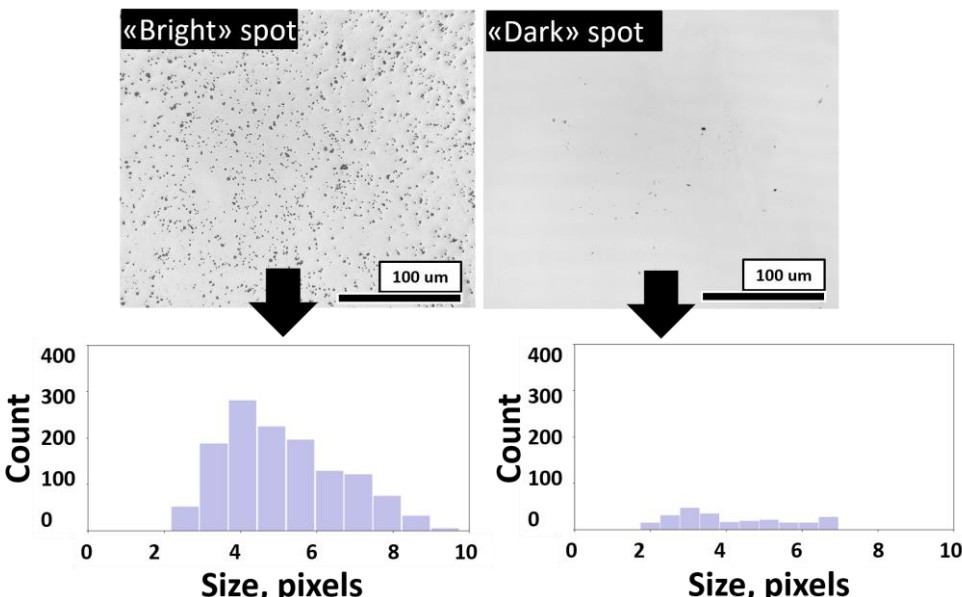

**Figure 20.** Top—laser confocal images of bright (**left**) and dark (**right**) spots after 60 min of polishing. Bottom—calculated distribution of abrasives for bright (**left**) and dark (**right**) spots.

### 3.3.3. Final Surface Analysis of Large Disk

Two sets of the disks were lapped using both larger (9 μm) and smaller (6 μm) diamond particles, followed by a polishing step with silica colloids lasting 120 min. The surface of the polycrystalline niobium disk has been checked with a laser confocal microscope at three different radii (see Figure 21). Each region corresponds to the total size of $1290 \times 1740$ μm$^2$ at six different positions measured along the corresponding radius with the dimensions of the scan of $290 \times 215$ μm$^2$. Comparing the average surface roughness values after polishing the lapped surfaces (see Figure 22), higher values were determined for the 6 μm diamond pretreated surface. The smaller diamond size of 6 μm provides a more uniform and consistent abrasive action during the pretreatment process, resulting in a more homogeneous final surface than after 9 μm. Moreover, this homogeneity results in a higher level of recrystallization during the polishing process, which leads to higher surface roughness values (Sa = 200 ± 50 nm) compared to bigger abrasives (Sa = 145 ± 40 nm).

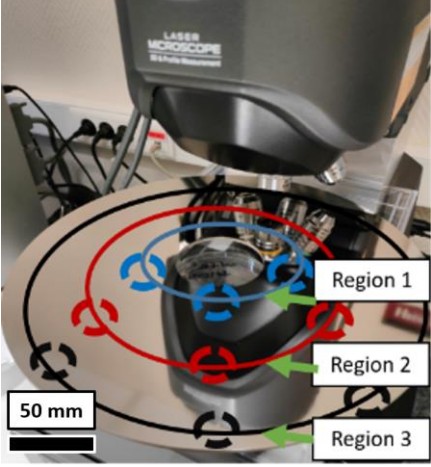

**Figure 21.** Photography of polished Nb disk with indicated regions which were analyzed with a laser confocal microscope. Note: polishing time 2 h.

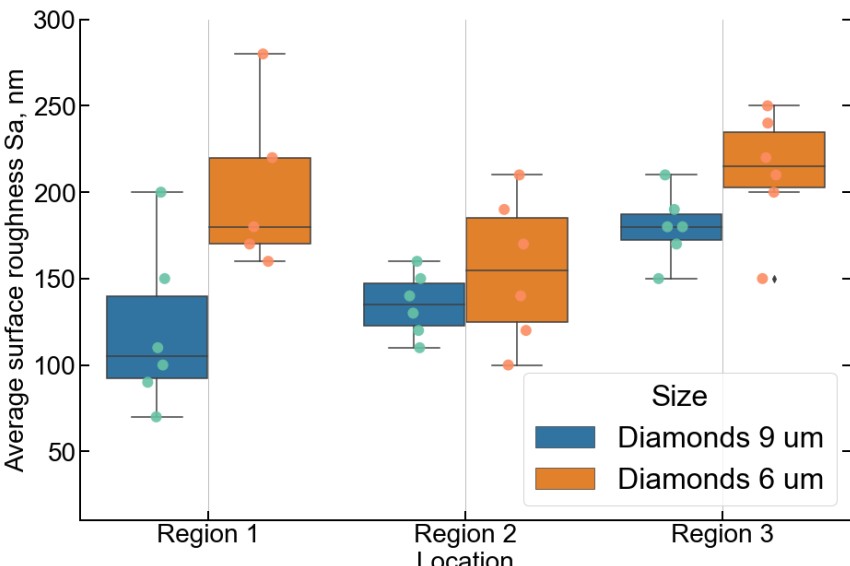

**Figure 22.** Measured surface roughness at three different radii. Region 1—inner edge, Region 2—center, and Region 3—outer edge.

Pictures in Figure 23 confirm the efficiency of Step 2 to remove embedded particles over the whole surface. However, following the execution of Step 2 on the disks that were previously lapped with 9 μm diamonds, visible scratches were apparent as depicted in Figure 23a–c. This phenomenon was not observed when utilizing smaller abrasives, such as 6 μm diamonds, see Figure 23d–f. Hence, a lapping step could be performed as a two-step process, which involves: (1) coarse treatment using 9 μm diamond particles, followed by (2) light lapping with 6 μm particles. This additional substep has the potential to reduce the time required to remove the damaged layer and improve the homogeneity of the surface prior to the final polishing step with a suitable slurry. Furthermore, the same patented disk could be used for all these sub-steps, resulting in fewer-manipulations and a more cost-effective process. At the sight of disk dimension, no advanced surface analysis for small samples could be performed. It is not expected that any major differences will arise as polishing parameters and consumables used are very similar to the ones used for small samples.

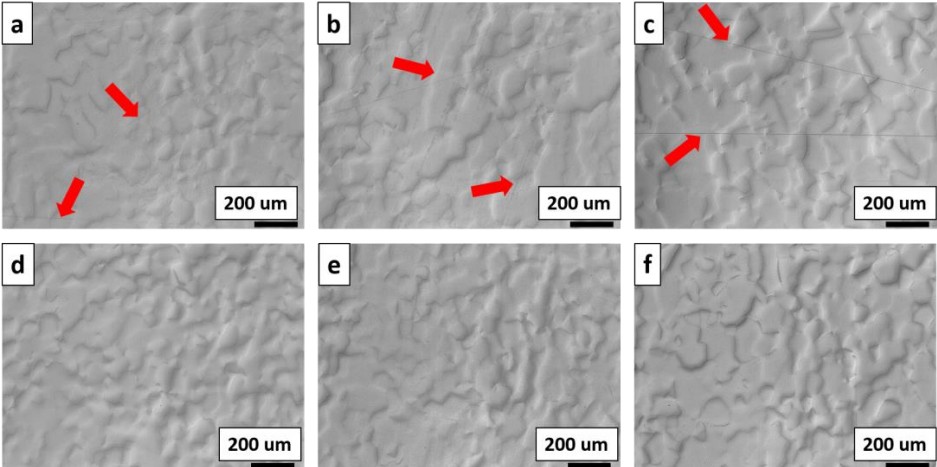

**Figure 23.** Photographs of the niobium surface after a full MP process at three different radii. Note: (**a**) Region 1—inner edge, (**b**) Region 2—center, and (**c**) Region 3—outer edge for Step 1 with 9 μm, followed by Step 2 for 120 min. (**d**) Region 1—inner edge, (**e**) Region 2—center, and (**f**) Region 3—outer edge for Step 1 with 6 μm, followed by Step 2 for 120 min. Note: Red arrows indicate the location of the scratches.

## 4. Conclusions

An alternative and simple 2-step polishing process was developed to first, efficiently remove the damaged layer and, secondly, improve the surface quality of niobium to meet the stringent requirements for superconducting accelerating cavities. Not only roughness but also surface purity and crystalline structure are of first importance to ensure good superconducting properties. The technique was first developed on small samples. Extensive surface analysis at room temperature as SIMS, XPS, and EBSD were performed and a very acceptable surface quality was demonstrated. Additional tests at cryogenic temperatures will be performed to evaluate the surface resistance under radiofrequency waves. The availability of such devices capable of resolving residual resistances as low as a few nano-Ohms is very low. This work is still in progress and results will be published in a future paper.

The compatibility of this technique for large disks has then been demonstrated. The procedure uses non-fixed diamonds between 3 μm and 9 μm depending on the sample dimensions, followed by a de-pollution step involving colloidal silica during at least 2 h to reach a good homogeneity all over the surface. The final roughness obtained, because of depollution requirements, is not optimal but is similar to the one achieved by EP. This technique is promising not only for bulk Nb, but also for substrate preparation (Nb, Cu . . . ) for future thin-film deposition. However, the fabrication of a complete 1.3 GHz single-cell cavity by this alternative path has to be demonstrated. Dedicated studies in collaboration with KEK are in progress.

**Author Contributions:** Conceptualization, C.Z.A. and D.L.; methodology and investigation, O.H., W.M., M.R., F.B. and S.G.; formal analysis, data curation, visualization, and writing original draft, O.H.; validation, supervision, D.L. and C.Z.A.; project administration, D.L.; funding acquisition, D.L., writing—review and editing, D.L., C.Z.A., F.B. and O.H. All authors have read and agreed to the published version of the manuscript.

**Funding:** Part of this work was supported by the European Nuclear Science and Application Research-2 (ENSAR-2) under grant agreement N° 654002.

**Data Availability Statement:** Not applicable.

**Acknowledgments:** We are very grateful and appreciate the support from the members of LAM PLAN team. Especially we thank W. Magnin and M. Rajkumar for the supporting assistance connected with the lapping and polishing tasks. Also, we would like to express our deepest thanks to S. Guilet (CNRS/INSP) for the invaluable assistance in the XPS analysis, and to F. Brisset (CNRS/IN2P3/ICMMO) for help with the EBSD measurements.

**Conflicts of Interest:** The authors declare no conflict of interest.

## Appendix A

Depth profile analysis was carried out based on the chosen masses with continuous sputtering by an Ar+ beam. In the oxide layer (see Figure A1), the signals of masses 27, 40, 44, and 109 are slightly higher in the case of the MP polished samples. At the moment when the oxide layer is over, the situation is contrary and signals are lower. It might be due to the fact that MP brings fewer hydrocarbons to the surface than conventional treatment.

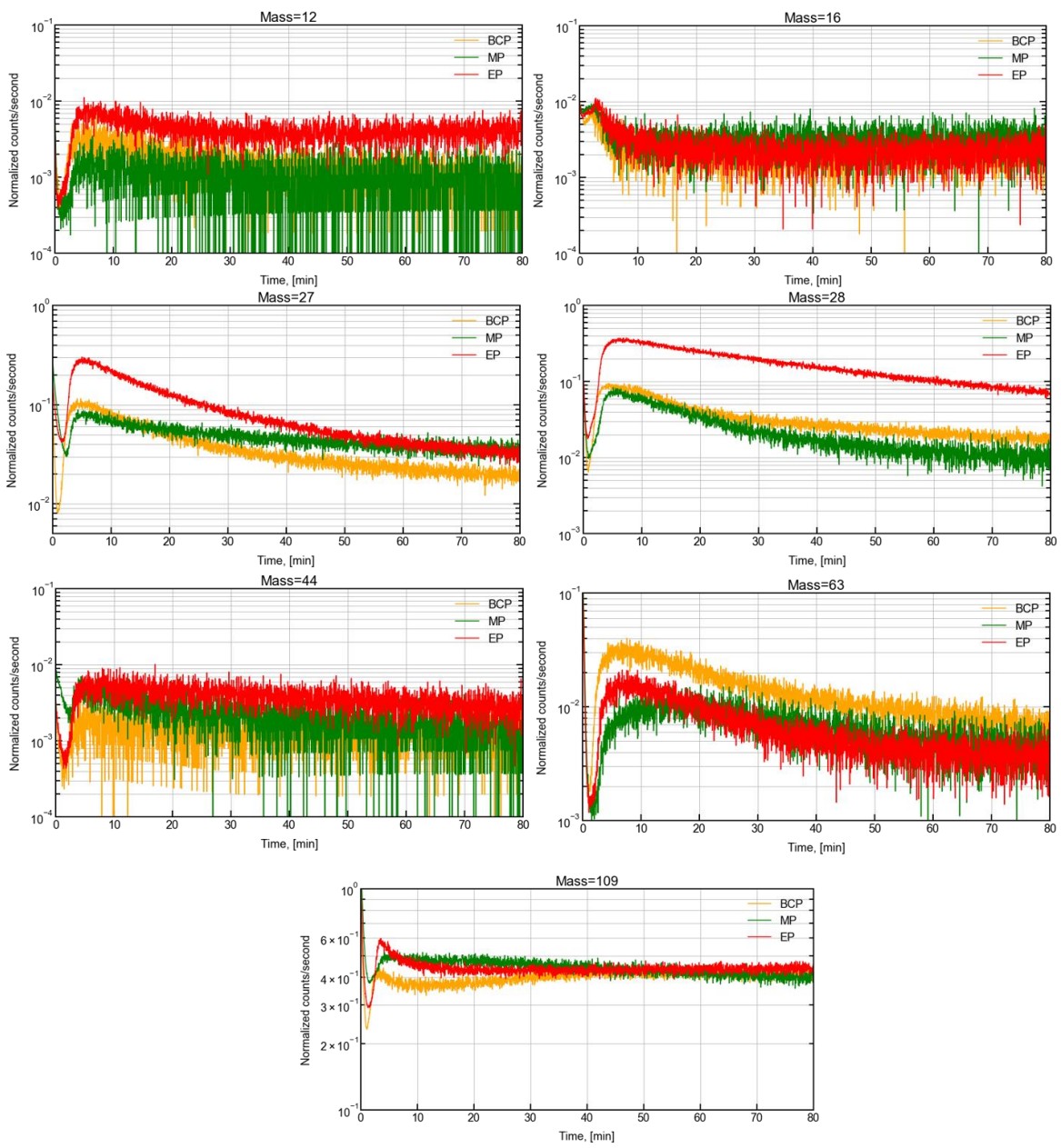

**Figure A1.** SIMS comparison of depth profiles after metallographic polishing (MP), buffered chemical polishing (BCP), and EP (electro-polishing) measured for desired investigated mass.

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
