# Peer review of "An Innovative Approach of Surface Polishing for SRF Cavity Applications"

_jmmp, doi:10.3390/jmmp7020062_

Round 1

Reviewer 1 Report

The manuscript deals with investigations into a new strategy for surface treatment of superconducting radio-frequence cavities. The results are of practical and scientific interest and the contents meet the focus of the journal. Before publication, however, the following amendments are considered necessary:

(1) The references in section “References” must be checked to ensure that all sources are listed in full. Internet sources for industrial websites should be removed and only scientific sources should be used. The source [43] is not referred to in the text.

(2) A number of abbreviations used in the text, captions and tables are not explained and must be briefly explained.

(3) It is suggested to name chapter 3 “Results and Discussion” and chapter 4 “Conclusions”. The last section (lines 525 ff.) seems to be misplaced.  

Author Response

Dear Reviewer,

I would like to express my gratitude for taking the time to read our manuscript and for providing valuable feedback and corrections.

I appreciate the thoroughness with which you reviewed my manuscript and the constructive criticism you offered, which has undoubtedly enhanced the clarity and coherence of my writing.

Once again, thank you for your helpful and relevant comments, and I integrated all corrections in the best possible way.

Please find an updated version attached to this message. This version is based on suggestions/corrections from several reviewers.

Best regards,

Oleksandr

Reviewer 2 Report

1. Please clearly and carefully revise the contents and explain consistently. If the main aim is to develop a “new” MP process and compare it with an established BCP process, please introduce both processes. Explain the “novelty” in contrast to applying a combination of existing processes, and then compare them. If EP represents another potential process to be compared, please also introduce and explain it with discrete parameters used.

2. The experimental procedures and results are both described in chapter 3 but should be more clearly separated. For example, the used samples and machines should be introduced in chapter 2.

3. The text on “SIMS and XPS Analysis” is quite confusing since the text and illustrations / tables are separated far from each other and therefore hardly retraceable. Please revise.

4. Please revise the whole manuscript using passive writing (such as “…was developed”) instead of active form (such as “we developed…”).

5. Please refer to the commented manuscript PDF for detailed corrections.

Author Response

Dear Reviewer,

I would like to express my gratitude for taking the time to read our manuscript and for providing valuable feedback and corrections.

I appreciate the thoroughness with which you reviewed my manuscript and the constructive criticism you offered, which has undoubtedly enhanced the clarity and coherence of my writing.

Once again, thank you for your helpful and relevant comments, and I integrated all corrections in the best possible way.

Please find an updated version attached to this message. This version is based on suggestions/corrections from several reviewers.

Reviewer 3 Report

Brief summary

This paper gives a proposition study for an advanced less hazardous large dimension surface polishing technique for Niobium in large scale accelerators to allow superconducting in a surface near area up to a depth of 100 nm and suppress local quenching and energy dissipation.

Broad comment

The amount of research and the consequent methodic design of surface treatment in connection with high standard analytical techniques is exemplary or even outstanding to my opinion. Thank you for sharing to this information to the community.

Introduction:

Line 30 and 33: Do you mean in less than 100 nm form the surface by “first 100s of nm”?

Results:

Line 420: please rewrite “Figure shows” in a way that results are shown in Figure…

Figure 25: I do not understand the indication “local defect” in the figure. Please add or explain as necessary.

Discussion:

The need to obtain effective superconductivity was well explained in the introduction but very short in the discussion. The authors state that conductivity measurements are ongoing and will be part of a future paper. To my opinion adding a closer discussion of the linkage of promising surface quality and possible elevated conductivity on the results could improve the take away merit of the reading.

Author Response

(The authors gave the same response as above.)

Round 2

Reviewer 2 Report

Dear authors,

many thanks for the very conscientious revision of the manuscript. In my opinion, the manuscript is now significantly improved in terms of presentation and textual contents. The comments have been carefully implemented. I therefore have no further corrections.